# *AutoGrow*: Automatic Layer Growing in Deep Convolutional Networks

## Abstract

Depth is a key component of Deep Neural Networks (DNNs), however, designing depth is heuristic and requires many human efforts. We propose *AutoGrow* to automate depth discovery in DNNs: starting from a shallow seed architecture, *AutoGrow* grows new layers if the growth improves the accuracy; otherwise, stops growing and thus discovers the depth. We propose robust growing and stopping policies to generalize to different network architectures and datasets. Our experiments show that by applying the same policy to different network architectures, *AutoGrow* can always discover near-optimal depth on various datasets of MNIST, FashionMNIST, SVHN, CIFAR10, CIFAR100 and ImageNet. For example, in terms of accuracy-computation trade-off, *AutoGrow* discovers a better depth combination in *ResNets* than human experts. Our *AutoGrow* is efficient. It discovers depth within similar time of training a single DNN.

## 1 Introduction

Layer depth is one of the decisive factors of the success of Deep Neural Networks (DNNs). For example, image classification accuracy keeps improving as the depth of network models grows (Krizhevsky et al., 2012; Simonyan & Zisserman, 2014; Szegedy et al., 2015; He et al., 2016; Huang et al., 2017). Although shallow networks cannot ensure high accuracy, DNNs composed of too many layers may suffer from over-fitting and convergence difficulty in training. How to obtain the optimal depth for a DNN still remains mysterious. For instance, *ResNet*-152 (He et al., 2016) uses 3, 8, 36 and 3 residual blocks under output sizes of $56 \times 56$, $28 \times 28$, $14 \times 14$ and $7 \times 7$, respectively, which don't show an obvious quantitative relation. In practice, people usually reply on some heuristic trials and tests to obtain the depth of a network: they first design a DNN with a specific depth and then train and evaluate the network on a given dataset; finally, they change the depth and repeat the procedure until the accuracy meets the requirement. Besides the high computational cost induced by the iteration process, such trial & test iterations must be repeated whenever dataset changes. In this paper, we propose *AutoGrow* that can automate depth discovery given a layer architecture. We will show that *AutoGrow* generalizes to different datasets and layer architectures.

There are some previous works which add or morph layers to increase the depth in DNNs. VggNet (Simonyan & Zisserman, 2014) and DropIn (Smith et al., 2016) added new layers into shallower DNNs; Network Morphism (Wei et al., 2016; 2017; Chen et al., 2015) morphed each layer to multiple layers to increase the depth meanwhile preserving the function of the shallower net. Table 1 summarizes differences in this work. Their goal was to overcome difficulty of training deeper DNNs or accelerate it. Our goal is to automatically find an optimal depth. Moreover, previous works applied layer growth by once or a few times at pre-defined locations to grow a pre-defined number of layers; in contrast, ours automatically learns the number of new layers and growth locations without limiting growing times. We will summarize more related works in Section 4.

Figure 1 illustrates an example of *AutoGrow*. It starts from the shallowest backbone network and gradually grows *sub-modules* (A *sub-module* can be one or more layers, *e.g.*, a residual block); the growth stops once a stopping policy is satisfied. We studied multiple initializers of new layers and multiple growing policies, and surprisingly find that: (1) a random initializer works equally or better than complicated Network Morphism; (2) it is more effective to grow *before* a shallow net converges. We hypothesize that this is because a converged shallow net is an inadequate initialization for training deeper net, while random initialization can help to escape from a bad starting point.

Motivated by this, we intentionally avoid full convergence during the growing by using (1) random initialization of new layers, (2) a constant large learning rate, and (3) a short growing interval.

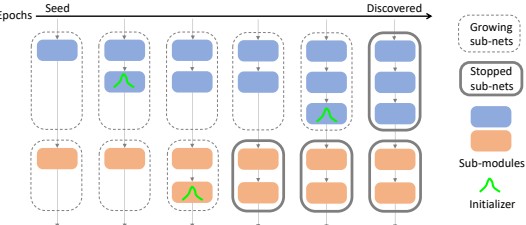

Figure 1: A simple example of *AutoGrow*.

|  | Previous works | Ours |
|---|---|---|
| Goal | Ease training | Depth automation |
| Times | Once or a few | Unlimited |
| Locations | Human defined | Learned |
| Layer # | Human defined | Learned |

Table 1: Comparison with previous works about layer growth.

Our contributions are: (1) We propose *AutoGrow* to automate DNN layer growing and depth discovery. *AutoGrow* is very robust. With the same hyper-parameters, it adapts network depth to various datasets including MNIST, FashionMNIST, SVHN, CIFAR10, CIFAR100 and ImageNet. Moreover, *AutoGrow* can also discover shallower DNNs when the dataset is a subset. (2) *AutoGrow* demonstrates high efficiency and scales up to ImageNet, because the layer growing is as fast as training a single DNN. On ImageNet, it discovers a new *ResNets* with better trade-off between accuracy and computation complexity. (3) We challenge the idea of Network Morphism, as random initialization works equally or better when growing layers. (4) We find that it is beneficial to rapidly grow layers before a shallower net converge, contradicting previous intuition.

## 2  *AutoGrow* – A DEPTH GROWING ALGORITHM

---
**Algorithm 1** *AutoGrow* Algorithm.

---
**Input**  :

1  A seed shallow network $g(\mathcal{X}_0)$ composed of $M$ sub-networks $\mathbb{F} = \{\boldsymbol{f}_i\,(\cdot;\mathbb{W}_i) : i = 0 \ldots M - 1\}$, where each sub-network has only one sub-module (a dimension reduction sub-module); an epoch interval $K$ to check growing and stopping policies; the number of fine-tuning epochs $N$ after growing.

**Initialization:**

2  A Circular Linked List of sub-networks under growing: `subNetList` $= \boldsymbol{f}_0\,(\cdot;\mathbb{W}_0) \to \cdots \to \boldsymbol{f}_{M-1}\,(\cdot;\mathbb{W}_{M-1})$;

3  The current growing sub-network: `growingSubNet` = `subNetList.head()` $= \boldsymbol{f}_0\,(\cdot;\mathbb{W}_0)$;

4  The recent grown sub-network: `grownSubNet` = `None`;

**Process**  :

5      *# if there exist growing sub-network(s)*

6      **while** `subNetList.size()>0` **do**

7          `train(`$g(\mathcal{X}_0)$`, K)`  *# train the whole network $g(\mathcal{X}_0)$ for K epochs*

8          **if** `meetStoppingPolicy()` **then**

9              *# remove a sub-network from the growing list if its growth did not improve accuracy*

10              `subNetList.delete(grownSubNet)`;

11          **end**

12          **if** `meetGrowingPolicy()` `and` `subNetList.size()>0` **then**

13              *# current growing sub-network* `growingSubNet` $== \boldsymbol{f}_i\,(\cdot;\mathbb{W}_i)$

14              $\mathbb{W}_i = \mathbb{W}_i \cup \mathcal{W}$  *# stack a sub-module on top of $\boldsymbol{f}_i\,(\cdot;\mathbb{W}_i)$*

15              `initializer(`$\mathcal{W}$`)`;  *# initialize the new sub-module $\mathcal{W}$*

16              *# record the recent grown sub-network and iterate to a next sub-network*

17              `grownSubNet` = `growingSubNet`;

18              `growingSubNet` = `subNetList.next(growingSubNet)`;

19          **end**

20      **end**

21  Fine-tune the discovered network $g(\mathcal{X}_0)$ for $N$ epochs;

**Output**  :

22  A trained neural network $g(\mathcal{X}_0)$ with learned depth.

---

Figure 1 gives an overview of the proposed *AutoGrow*. In this paper, we use *network*, *sub-networks*, *sub-modules* and *layers* to describe the architecture hierarchy. A *network* is composed of a cascade of *sub-networks*. A *sub-network* is composed of *sub-modules*, which typical share the same output size. A *sub-module* (*e.g.* a residual block) is an elementary growing block composed of one or a few *layers*. In this section, we rigorously formulate a generic version of *AutoGrow* which will be materialized in subsections. A deep convolutional *network* $g(\mathcal{X}_0)$ is a cascade of *sub-networks* by composing functions as $g(\mathcal{X}_0) = l\left(\boldsymbol{f}_{M-1}\left(\boldsymbol{f}_{M-2}\left(\cdots\boldsymbol{f}_1\left(\boldsymbol{f}_0\left(\mathcal{X}_0\right)\right)\cdots\right)\right)\right)$, where $\mathcal{X}_0$ is an input image, $M$ is the number of sub-networks, $l(\cdot)$ is a loss function, and $\mathcal{X}_{i+1} = \boldsymbol{f}_i\left(\mathcal{X}_i\right)$ is a *sub-network* that operates on an input image or a feature tensor $\mathcal{X}_i \in \mathbb{R}^{c_i \times h_i \times w_i}$. Here, $c_i$ is the number of channels, and $h_i$ and $w_i$ are spatial dimensions. $\boldsymbol{f}_i\left(\mathcal{X}_i\right)$ is a simplified notation of $\boldsymbol{f}_i\left(\mathcal{X}_i;\mathbb{W}_i\right)$, where $\mathbb{W}_i$

is a set of *sub-modules*' parameters within the $i$-th *sub-network*. Thus $\mathbb{W} = \{\mathbb{W}_i : i = 0 \ldots M - 1\}$ denotes the whole set of parameters in the DNN. To facilitate growing, the following properties are supported within a sub-network: (1) the first sub-module usually reduces the size of input feature maps, *e.g.*, using pooling or convolution with a stride; and (2) all sub-modules in a sub-network maintain the same output size. As such, our framework can support popular networks, including *VggNet*-like plain networks (Simonyan & Zisserman, 2014), *GoogLeNet* (Szegedy et al., 2015), *ResNets* (He et al., 2016) and *DenseNets* (Huang et al., 2017). In this paper, we select *ResNets* and *VggNet*-like nets as representatives of DNNs with and without shortcuts, respectively.

With above notations, Algorithm 1 rigorously describes the *AutoGrow* algorithm. In brief, *AutoGrow* starts with the shallowest net where every sub-network has only one sub-module for spatial dimension reduction. *AutoGrow* loops over all growing sub-networks in order. For each sub-network, *AutoGrow* stacks a new sub-module. When the new sub-module does not improve the accuracy, the growth in corresponding sub-network will be permanently stopped. The details of our method will be materialized in the following subsections.

## 2.1 SEED SHALLOW NETWORKS AND SUB-MODULES

In this paper, in all datasets except ImageNet, we explore growing depth for four types of DNNs: **(1)** `Basic3ResNet`: the same *ResNet* used for CIFAR10 in He et al. (2016), which has 3 residual *sub-networks* with output spatial sizes of $32 \times 32$, $16 \times 16$ and $8 \times 8$, respectively; **(2)** `Basic4ResNet`: a variant of *ResNet* used for ImageNet in He et al. (2016) built by basic residual blocks (each of which contains two convolutions and one shortcut). There are 4 *sub-networks* with output spatial sizes of $32 \times 32$, $16 \times 16$, $8 \times 8$ and $4 \times 4$, respectively; **(3)** `Plain3Net`: a *VggNet*-like plain net by removing shortcuts in `Basic3ResNet`; **(4)** `Plain4Net`: a *VggNet*-like plain net by removing shortcuts in `Basic4ResNet`.

In *AutoGrow*, the architectures of seed shallow networks and sub-modules are pre-defined. In plain DNNs, a *sub-module* is a stack of convolution, Batch Normalization and ReLU; in residual DNNs, a *sub-module* is a residual block. In *AutoGrow*, a *sub-network* is a stack of all sub-modules with the same output spatial size. Unlike He et al. (2016) which manually designed the depth, *AutoGrow* starts from a seed architecture in which each sub-network has only one sub-module and automatically learns the number of sub-modules.

On ImageNet, we apply the same backbones in He et al. (2016) as the seed architectures. A seed architecture has only one sub-module under each output spatial size. For a *ResNet* using basic residual blocks or bottleneck residual blocks (He et al., 2016), we respectively name it as `Basic4ResNet` or `Bottleneck4ResNet`. `Plain4Net` is also obtained by removing shortcuts in `Basic4ResNet`.

## 2.2 SUB-MODULE INITIALIZERS

Here we explain how to initialize a new sub-module $\mathcal{W}$ in `initializer(`$\mathcal{W}$`)` mentioned in Algorithm 1. Network Morphism changes DNN architecture meanwhile preserving the loss function via special initialization of new layers, that is, $g(\mathcal{X}_0; \mathbb{W}) = g(\mathcal{X}_0; \mathbb{W} \cup \mathcal{W}) \; \forall \mathcal{X}_0$. A residual sub-module shows a nice property: when stacking a residual block and initializing the last Batch Normalization layer as zeros, the function of the shallower net is preserved but the DNN is morphed to a deeper net. Thus, Network Morphism can be easily implemented by this zero initialization (`ZeroInit`).

In this work, all layers in $\mathcal{W}$ are initialized using default randomization, except for a special treatment of the last Batch Normalization layer in a *residual* sub-module. Besides `ZeroInit`, we propose a new `AdamInit` for Network Morphism. In `AdamInit`, we freeze all parameters except the last Batch Normalization layer in $\mathcal{W}$, and then use Adam optimizer (Kingma & Ba, 2014) to optimize the last Bath Normalization for maximum 10 epochs till the training accuracy of the deeper net is as good as the shallower one. After `AdamInit`, all parameters are jointly optimized. We view `AdamInit` as a Network Morphism because the training loss is similar after `AdamInit`. We empirically find that `AdamInit` can usually find a solution in less than 3 epochs. We also study random initialization of the last Batch Normalization layer using uniform (`UniInit`) or Gaussian (`GauInit`) noises with a standard deviation 1.0. We will show that `GauInit` obtains the best result, challenging the idea of Network Morphism (Wei et al., 2016; 2017; Chen et al., 2015).

## 2.3 GROWING AND STOPPING POLICIES

In Algorithm 1, a growing policy refers to `meetGrowingPolicy()`, which returns true when the network should grow a sub-module. Two growing policies are studied here:

1. Convergent Growth: `meetGrowingPolicy()` returns true when the improvement of validation accuracy is less than $\tau$ in the last $K$ epochs. That is, in Convergent Growth, *AutoGrow* only grows when current network has converged. This is a similar growing criterion adopted in previous works (Elsken et al., 2017; Cai et al., 2018a;b).
2. Periodic Growth: `meetGrowingPolicy()` always returns true, that is, the network always grows every $K$ epochs. Therefore, $K$ is also the growing period. In the best practice of *AutoGrow*, $K$ is small (*e.g.* $K = 3$) such that it grows before current network converges.

Our experiments will show that Periodic Growth outperforms Convergent Growth. We hypothesize that a fully converged shallower net is an inadequate initialization to train a deeper net. We will perform experiments to test this hypothesis and visualize optimization trajectory to illustrate it.

A stopping policy denotes `meetStoppingPolicy()` in Algorithm 1. When Convergent Growth is adopted, `meetStoppingPolicy()` returns true if a recent growth does not improve validation accuracy more than $\tau$ within $K$ epochs. We use a similar stopping policy for Periodic Growth; however, as it can grow rapidly with a small period $K$ (*e.g.* $K = 3$) before it converges, we use a larger window size $J$ for stop. Specifically, when Periodic Growth is adopted, `meetStoppingPolicy()` returns true when the validation accuracy improves less than $\tau$ in the last $J$ epochs, where $J \gg K$.

Hyper-parameters $\tau$, $J$ and $K$ control the operation of *AutoGrow* and can be easily setup and generalize well. $\tau$ denotes the significance of accuracy improvement for classification. We simply set $\tau = 0.05\%$ in all experiments. $J$ represents how many epochs to wait for an accuracy improvement before stopping the growth of a sub-network. It is more meaningful to consider stopping when the new net is trained to some extent. As such, we set $J$ to the number of epochs $T$ under the largest learning rate when training a baseline. $K$ means how frequently *AutoGrow* checks the polices. In Convergent Growth, we simply set $K = T$, which is long enough to ensure convergence. In Periodic Growth, $K$ is set to a small fraction of $T$ to enable fast growth before convergence; more importantly, $K = 3$ is very robust to all networks and datasets. Therefore, all those hyper-parameters are very robust and strongly correlated to design considerations.

## 3 EXPERIMENTS

In this paper, we use `Basic3ResNet-2-3-2`, for instance, to denote a model architecture which contains 2, 3 and 2 sub-modules in the first, second and third sub-networks, respectively. Sometimes we simplify it as `2-3-2` for convenience. *AutoGrow* always starts from the shallowest depth of `1-1-1` and uses the maximum validation accuracy as the metric to guide growing and stopping. All DNN baselines are trained by SGD with momentum 0.9 using staircase learning rate. The initial learning rate is 0.1 in *ResNets* and 0.01 in plain networks. On ImageNet, baselines are trained using batch size 256 for 90 epochs, within which learning rate is decayed by $0.1\times$ at epoch 30 and 60. In all other smaller datasets, baselines are trained using batch size 128 for 200 epochs and learning rate is decayed by $0.1\times$ at epoch 100 and 150.

Our early experiments followed prior wisdom by growing layers with Network Morphism (Wei et al., 2016; 2017; Chen et al., 2015; Elsken et al., 2017; Cai et al., 2018a;b), *i.e.*, *AutoGrow* with `ZeroInit` (or `AdamInit`) and Convergent Growth policy; however, it stopped early with very shallow DNNs, failing to find optimal depth. We hypothesize that a converged shallow net with Network Morphism gives a bad initialization to train a deeper neural network. Section 3.1 experimentally test that the hypothesis is valid. To tackle this issue, we intentionally avoid convergence during growing by three simple solutions, which are evaluated in Section 3.2. Finally, Section 3.3 and Section 3.4 include extensive experiments to show the effectiveness of our final *AutoGrow*.

## 3.1 Suboptimum of Network Morphism and Convergent Growth

In this section, we study Network Morphism itself and its integration into our *AutoGrow* under Convergent Growth. When studying Network Morphism, we take the following steps: 1) train a shallower *ResNet* to converge, 2) stack residual blocks on top of each sub-network to morph to a deeper net, 3) use `ZeroInit` or `AdamInit` to initialize new layers, and 4) train the deeper net in a standard way. We compare the accuracy difference ("$\Delta$") between Network Morphism and training the deeper net from scratch. Table 2 summaries our results. Network Morphism has a lower accuracy (negative "$\Delta$") in all the cases, which validates our hypothesis that a converged shallow network with Network Morphism gives a bad initialization to train a deeper net. We visualize the optimization trajectories in Appendix A.0.1 to illustrate the hypothesis.

To further validate our hypothesis, we integrate Network Morphism as the initializer in *AutoGrow* with Convergent Growth policy. We refer to this version of *AutoGrow* as *c-AutoGrow* with "*c-*" denoting "Convergent." More specific, we take `ZeroInit` or `AdamInit` as sub-module initializer and "Convergent Growth" policy in Algorithm 1. To recap, in this setting, *AutoGrow* trains a shallower net till it converges, then grows a sub-module which is initialized by Network Morphism, and repeats the same process till there is no further accuracy improvement. In every interval of $K$ training epochs ($\mathtt{train}\,(g(\mathcal{X}_0), K)$ in Algorithm 1), "staircase" learning rate is used. The learning rate is reset to $0.1$ at the first epoch, and decayed by $0.1\times$ at epoch $\frac{K}{2}$ and $\frac{3K}{4}$. The results are shown in Table 3 by "*staircase*" rows, which illustrate that *c-AutoGrow* can grow a DNN multiple times and finally find a depth. However, there are two problems: 1) the final accuracy is lower than training the found net from scratch, as indicated by "$\Delta$", validating our hypothesis; 2) the depth learning stops too early with a relatively shallower net, while a deeper net beyond the found depth can achieve a higher accuracy as we will show in Table 6. These problems provide a circumstantial evidence of the hypothesis that a converged shallow net with Network Morphism gives a bad initialization. Thus, *AutoGrow* cannot receive signals to continue growing after a limited number of growths. In Appendix A.0.1, Figure 6(a) visualizes the trajectory of *c-AutoGrow* corresponding to row "2-3-6" in Table 3.

## 3.2 Ablation Study for *AutoGrow* Design

Based on the findings in Section 3.1, we propose three simple but effective solutions to further enhance *AutoGrow* and refer it as *p-AutoGrow*, with "*p-*" denoting "Periodic": **(1)** Use a *large constant* learning rate for growing, *i.e.*, $0.1$ for residual networks and $0.01$ for plain networks. Stochastic gradient descent with a large learning rate intrinsically introduces noises, which help to avoid a full convergence into a bad initialization from a shallower net. Note that staircase learning rate is still used for fine-tuning after discovering the final DNN; **(2)** Use *random* initialization (`UniInit` or `GauInit`) as noises to escape from an inadequate initialization; **(3)** Grow rapidly *before* a shallower net converges by taking Periodic Growth with a small $K$.

*p-AutoGrow* is our final *AutoGrow*. In the rest part of this section, we perform ablation study to prove that the three solutions are effective. We start from *c-AutoGrow*, and incrementally add above solutions one by one and eventually obtain *p-AutoGrow*. In Table 3, first, we replace the staircase learning rate with a constant learning rate, the accuracy of *AutoGrow* improves and therefore "$\Delta$" improves; second, we further replace Network Morphism (`ZeroInit` or `AdamInit`) with a random initializer (`UniInit` or `GauInit`) and result in a bigger gain. Overall, combining a constant learning rate with `GauInit` performs the best. Thus, constant learning rate and `GauInit` are adopted in the remaining experiments, unless we explicitly specify them.

Table 2: Network Morphism tested on CIFAR10.

| net backbone | shallower | deeper | initializer | accu % | $\Delta^*$ |
|---|---|---|---|---|---|
| Basic3ResNet | 3-3-3 | 5-5-5 | ZeroInit | 92.71 | -0.77 |
| | | | AdamInit | 92.82 | -0.66 |
| Basic3ResNet | 5-5-5 | 9-9-9 | ZeroInit | 93.64 | -0.27 |
| | | | AdamInit | 93.53 | -0.38 |
| Basic4ResNet | 1-1-1-1 | 2-2-2-2 | ZeroInit | 94.96 | -0.37 |
| | | | AdamInit | 95.17 | -0.16 |

\* $\Delta =$ (accuracy of Network Morphism) $-$ (accuracy of training from scratch)

Table 3: Ablation study of *c-AutoGrow*.

| dataset | learning rate | initializer | found net† | accu % | Δ* | dataset | learning rate | initializer | found net† | accu % | Δ* |
|---------|---------------|-------------|------------|--------|-----|---------|---------------|-------------|------------|--------|-----|
| | *staircase* | ZeroInit | 2-3-6 | 91.77 | -1.06 | | *staircase* | ZeroInit | 4-3-4 | 70.04 | -0.65 |
| | *staircase* | AdamInit | 3-4-3 | 92.21 | -0.59 | | *staircase* | AdamInit | 3-3-3 | 69.85 | -0.65 |
| CIFAR10 | constant | ZeroInit | 2-2-4 | 92.23 | 0.16 | CIFAR100 | constant | ZeroInit | 3-2-4 | 70.22 | 0.35 |
| | constant | AdamInit | 3-4-4 | 92.60 | -0.41 | | constant | AdamInit | 3-3-3 | 70.00 | -0.50 |
| | constant | UniInit | 3-4-4 | 92.93 | -0.08 | | constant | UniInit | 4-4-3 | 70.39 | 0.36 |
| | **constant** | **GauInit** | **2-4-3** | **93.12** | **0.55** | | **constant** | **GauInit** | **3-4-3** | **70.66** | **0.91** |

† Basic3ResNet      * Δ = (accuracy of *c-AutoGrow*) − (accuracy of training from scratch)

Note that, in this paper, we are more interested in automating depth discovery to find a final DNN ("found net") with a high accuracy ("accu"). Ideally, the "found net" has a minimum depth, a larger depth than which cannot further improve "accu". We will show in Figure 3 that *AutoGrow* discovers a depth approximately satisfying this property. The "Δ" is a metric to indicate how well shallower nets initialize deeper nets; a negative "Δ" indicates that weight initialization from a shallower net hurts training of a deeper net; while a positive "Δ" indicates *AutoGrow* helps training a deeper net, which is a byproduct of this work.

Finally, we apply the last solution – Periodic Growth, and obtain our final *p-AutoGrow*. Our ablation study results for *p-AutoGrow* are summarized in Table 5 and Table 4. Table 5 analyzes the impact of the growing period $K$. In general, $K$ is a hyper-parameter to trade off speed and accuracy: a smaller $K$ takes a longer learning time but discovers a deeper net, vice versa. Our results validate the preference of a faster growth (*i.e.* a smaller $K$). On CIFAR10/CIFAR100, the accuracy reaches plateau/peak at $K = 3$; further reducing $K$ produces a deeper net while the accuracy gain is marginal/impossible. In the following, we simply select $K = 3$ for robustness test. More importantly, our quantitative results in Table 5 show that *p-AutoGrow* finds much deeper nets, overcoming the very-early stop issue in *c-AutoGrow* in Table 3. That is, Periodic Growth proposed in this work is much more effective than Convergent Growth utilized in previous work.

For sanity check, we perform the ablation study of initializers for *p-AutoGrow*. The results are in Table 8 in Appendix A.0.3, which further validates our wisdom on selecting GauInit. The motivation of Network Morphism in previous work was to start a deeper net from a loss function that has been well optimized by a shallower net, so as not to restart the deeper net training from scratch (Wei et al., 2016; 2017; Chen et al., 2015; Elsken et al., 2017; Cai et al., 2018a;b). In all our experiments, we find this is sure even with random initialization. Figure 2 plots the convergence curves and learning process for "42-42-42" in Table 5. Even with GauInit, the loss and accuracy rapidly recover and no restart is observed. The convergence pattern in the "Growing" stage is similar to the "Fine-tuning" stage under the same learning rate (the initial learning rate 0.1). Similar results on ImageNet will be shown in Figure 8. Our results challenge the necessity of Network Morphism when growing neural networks.

At last, we perform the ablation study on the initial depth of the seed network. Table 4 demonstrates that a shallowest DNN works as well as a deeper seed. This implies that *AutoGrow* can appropriately stop regardless of the depth of the seed network. As the focus of this work is on depth automation, we prefer starting with the shallowest seed to avoid a manual search of a seed depth.

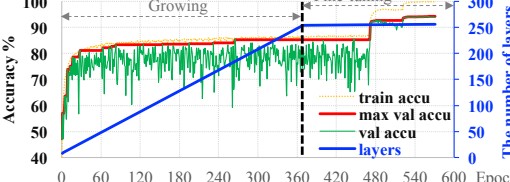

Figure 2: *p-AutoGrow* on CIFAR10 ($K = 3$). The seed net is Basic3ResNet-1-1-1.

Table 4: *p-AutoGrow* with different seed architecture.

| dataset | seed net† | found net† | accuracy % |
|---------|-----------|------------|------------|
| CIFAR10 | 1-1-1 | 42-42-42 | 94.27 |
| | 5-5-5 | 46-46-46 | 94.16 |
| CIFAR10 | 1-1-1-1 | 22-22-22-22 | 95.49 |
| | 5-5-5-5 | 23-22-22-22 | 95.62 |

† Basic3ResNet or Basic4ResNet.

### 3.3 ADAPTABILITY OF *AutoGrow*

To verify the adaptability of *AutoGrow*, we use an identical configuration (*p-AutoGrow* with $K = 3$) and test over 5 datasets and 4 seed architectures. Table 6 includes the results of all 20 combinations. Figure 3 compares *AutoGrow* with manual search which is obtained by training many DNNs with different depths from scratch. The results lead to the following conclusions and contributions:

Table 5: *p-AutoGrow* with different growing interval $K$.

| | CIFAR10 | | | CIFAR100 | |
|---|---|---|---|---|---|
| $K$ | found net[†] | accu % | $K$ | found net[†] | accu % |
| 50 | 6-5-3 | 92.95 | 50 | 8-5-7 | 72.07 |
| 20 | 7-7-7 | 93.26 | 20 | 8-11-10 | 72.93 |
| 10 | 19-19-19 | 93.46 | 10 | 18-18-18 | 73.64 |
| 5 | 23-22-22 | 93.98 | 5 | 23-23-23 | 73.70 |
| **3** | **42-42-42** | **94.27** | **3** | **54-53-53** | **74.72** |
| 1 | 77-76-76 | 94.30 | 1 | 68-68-68 | 74.51 |
| [†] Basic3ResNet | | | [†] Basic3ResNet | | |

Table 6: The adaptability of *AutoGrow* to datasets

| net | dataset | found net | accu % | $\Delta^*$ | net | dataset | found net | accu % | $\Delta^*$ |
|---|---|---|---|---|---|---|---|---|---|
| | CIFAR10 | 42-42-42 | 94.27 | -0.03 | | CIFAR10 | 23-22-22 | 90.82 | **6.49** |
| | CIFAR100 | 54-53-53 | 74.72 | **-0.95** | | CIFAR100 | 28-28-27 | 66.34 | **31.53** |
| Basic3ResNet | SVHN | 34-34-34 | 97.22 | 0.04 | Plain3Net | SVHN | 36-35-35 | 96.79 | **77.20** |
| | FashionMNIST | 30-29-29 | 94.57 | -0.06 | | FashionMNIST | 17-17-17 | 94.49 | **0.56** |
| | MNIST | 33-33-33 | 99.64 | -0.03 | | MNIST | 20-20-20 | 99.66 | **0.12** |
| | CIFAR10 | 22-22-22-22 | 95.49 | -0.10 | | CIFAR10 | 17-17-17-17 | 94.20 | **5.72** |
| | CIFAR100 | 17-51-16-16 | 79.47 | **1.22** | | CIFAR100 | 16-15-15-15 | 73.91 | **29.34** |
| Basic4ResNet | SVHN | 20-20-19-19 | 97.32 | -0.08 | Plain4Net | SVHN | 12-12-12-11 | 97.08 | **0.32** |
| | FashionMNIST | 27-27-27-26 | 94.62 | **-0.17** | | FashionMNIST | 13-13-13-13 | 94.47 | **0.72** |
| | MNIST | 11-10-10-10 | 99.66 | 0.01 | | MNIST | 13-12-12-12 | 99.57 | 0.03 |

[*] $\Delta$ = (accuracy of *AutoGrow*) − (accuracy of training from scratch)

1. In Table 6, *AutoGrow* discovers layer depth across all scenarios without any tuning, achieving the main goal of this work. Manual design needs $m \cdot n \cdot k$ trials, where $m$ and $n$ are respectively the numbers of datasets and sub-module categories, and $k$ is the number of trials per dataset per sub-module category;

2. For *ResNets*, a discovered depth ("●" in Figure 3) falls at the location where accuracy saturates. This means *AutoGrow* discovers a near-optimal depth: a shallower depth will lose accuracy while a deeper one gains little. The final accuracy of *AutoGrow* is as good as training the discovered net from scratch as indicated by "$\Delta$" in Table 6, indicating that initialization from shallower nets does not hurt training of deeper nets. As a byproduct, in plain networks, there are large positive "$\Delta$"s in Table 6. It implies that baselines fail to train very deep plain networks even using Batch Normalization, but *AutoGrow* enables the training of these networks; In Appendix A.0.3, Table 9 shows the accuracy improvement of plain networks by tuning $K$, approaching the accuracy of *ResNets* with the same depth.

3. For robustness and generalization study purpose, we stick to $K = 3$ in our experiments, however, we can tune $K$ to trade off accuracy and model size. As shown in Figure 3, *AutoGrow* discovers smaller DNNs when increasing $K$ from 3 ("●") to 50 ("○"). Interestingly, the accuracy of plain networks even increases at $K = 50$. This implies the possibility of discovering a better accuracy-depth trade-off by tuning $K$, although we stick to $K = 3$ for generalizability study and it generalizes well.

4. In Table 6, *AutoGrow* discovers different depths under different sub-modules. The final accuracy is limited by the sub-module design, not by our *AutoGrow*. Given a sub-module architecture, our *AutoGrow* can always find a near-optimal depth. With a better sub-module architecture, such as NASNet cell (Zoph et al., 2018), *AutoGrow* can improve accuracy.

Finally, our supposition is that: when the size of dataset is smaller, the optimal depth should be smaller. Under this supposition, we test the effectiveness of *AutoGrow* by sampling a subset of dataset and verify if *AutoGrow* can discover a shallower depth. In Appendix A.0.3, Table 11 summarizes the results. As expected, our experiments show that *AutoGrow* adapts to shallower networks when the datasets are smaller.

## 3.4 SCALING TO IMAGENET AND EFFICIENCY

In ImageNet, $K = 3$ should generalize well, but we explore *AutoGrow* with $K = 2$ and $K = 5$ to obtain an accuracy-depth trade-off line for comparison with human experts. The larger $K = 5$ enables *AutoGrow* to obtain a smaller DNN to trade-off accuracy and model size (computation) and the smaller $K = 2$ achieves higher accuracy. The results are shown in Table 7, which proves that *AutoGrow* automatically finds a good depth without any tuning. As a byproduct, the accuracy is even higher than training the found net from scratch, indicating that the Periodic Growth in *AutoGrow*

Table 7: Scaling up to ImageNet

| net | $K$ | found net | Top-1 | Top-5 | [†]$\Delta$ Top-1 |
|---|---|---|---|---|---|
| `Basic4ResNet` | 2 | `12-12-11-11` | 76.28 | 92.79 | 0.43 |
| | 5 | `9-3-6-4` | 74.75 | 91.97 | 0.72 |
| `Bottleneck4ResNet` | 2 | `6-6-6-17` | 77.99 | 93.91 | 0.83 |
| | 5 | `6-7-3-9` | 77.33 | 93.65 | 0.83 |
| `Plain4Net` | 2 | `6-6-6-6` | 71.22 | 90.08 | 0.70 |
| | 5 | `5-5-5-4` | 70.54 | 89.76 | 0.93 |

[†] $\Delta$ = (Top-1 of *AutoGrow*) − (Top-1 of training from scratch)

helps training deeper nets. The comparison of *AutoGrow* and manual depth design (He et al., 2016) is in Figure 4, which shows that *AutoGrow* achieves better trade-off between accuracy and computation (measured by floating point operations).

In Appendix A.0.3, Table 10 summarizes the breakdown of wall-clock time in *AutoGrow*. The growing/searching time is as efficient as (often more efficient than) fine-tuning the single discovered DNN. The scalability of *AutoGrow* comes from its intrinsic features that (1) it grows quickly with a short period $K$ and stops immediately if no improvement is sensed; and (2) the network is small at the beginning of growing.

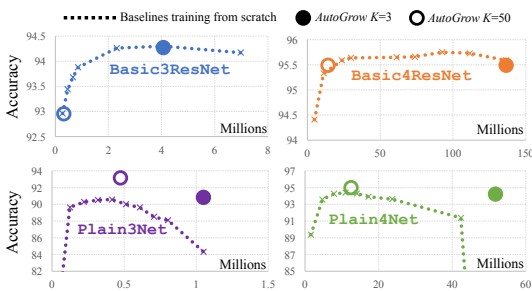

Figure 3: *AutoGrow* vs manual search obtained by training many baselines from scratch. $x-axis$ is the number of parameters. Dataset is CIFAR10.

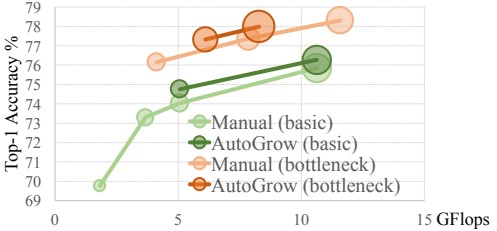

Figure 4: *AutoGrow vs.* manual design (He et al., 2016) on ImageNet. Marker area is proportional to model size determined by depth. "basic"("bottleneck") refers to *ResNets* with basic (bottleneck) residual blocks.

## 4 RELATED WORK

Neural Architecture Search (NAS) (Zoph & Le, 2016) and neural evolution (Miikkulainen et al., 2019; Angeline et al., 1994; Stanley & Miikkulainen, 2002; Liu et al., 2017a; Real et al., 2017) can search network architectures from a gigantic search space. In NAS, the depth of DNNs in the search space is fixed, while *AutoGrow* learns the depth. Some NAS methods (Bender et al., 2018; Liu et al., 2018b; Cortes et al., 2017) can find DNNs with different depths, however, the maximum depth is pre-defined and shallower nets are obtained by padding zero operations or selecting shallower branches, while our *AutoGrow* learns the depth in an open domain to find a minimum depth, beyond which no accuracy improvement can be obtained. Moreover, NAS is very computation and memory intensive. To accelerate NAS, one-shot models (Saxena & Verbeek, 2016; Pham et al., 2018; Bender et al., 2018), DARTS (Liu et al., 2018b) and NAS with Transferable Cell (Zoph et al., 2018; Liu et al., 2018a) were proposed. The search time reduces dramatically but is still long from practical perspective. It is still very challenging to deploy these methods to larger datasets such as ImageNet. In contrast, our *AutoGrow* can scale up to ImageNet thanks to its short depth learning time, which is as efficient as training a single DNN.

In addition to architecture search which requires to train lots of DNNs from scratch, there are also many studies on learning neural structures within a single training. Structure pruning and growing were proposed for different goals, such as efficient inference (Wen et al., 2016; Li et al., 2016; Lebedev & Lempitsky, 2016; He et al., 2017; Luo et al., 2017; Liu et al., 2017b; Dai et al., 2017; Huang et al., 2018; Gordon et al., 2018; Du et al., 2019), lifelong learning (Yoon et al., 2017) and model adaptation (Feng & Darrell, 2015; Philipp & Carbonell, 2017). However, those works fixed the network depth and limited structure learning within the existing layers. Optimization over a DNN with fixed depth is easier as the skeleton architecture is known. *AutoGrow* performs in a scenario where the DNN depth is unknown hence we need to seek for the optimal depth.

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

# A APPENDIX

### A.0.1 OPTIMIZATION TRAJECTORIES OF NETWORK MORPHISM

We hypothesize that a converged shallower net may not be an adequate initialization. Figure 5 visualizes and compares the optimization trajectories of Network Morphism and the training from scratch. In this figure, the shallower net is `Basic3ResNet-3-3-3` (*ResNet*-20) and the deeper one is `Basic3ResNet-5-5-5` (*ResNet*-32) in Table 2. The initializer is `ZeroInit`. The visualization method is extended from Li et al. (2018). Points on the trajectory are evenly sampled every a few epochs. To maximize the variance of trajectory, we use PCA to project from a high dimensional space to a 2D space and use the first two Principle Components (PC) to form the axes in Figure 5. The contours of training loss function and the trajectory are visualized around the final minimum of the deeper net. When projecting a shallower net to a deeper net space, zeros are padded for the parameters not existing in the deeper net. We must note that the loss increase along the trajectory does not truly represent the situation in high dimensional space, as the trajectory is just a projection. It is possible that the loss remains decreasing in the high dimension while it appears in an opposite way in the 2D space. The sharp detour at "Morphing" in Figure 5(a) may indicate that the shallower net plausibly converges to a point that the deeper net struggles to escape. In contrast, Figure 5(b) shows that the trajectory of the direct optimization in the deeper space smoothly converges to a better minimum.

Figure 6(a) visualizes the trajectory of *c-AutoGrow* corresponding to row "2-3-6" in Table 3. Along the trajectory, there are many trials to detour and escape an initialization from a shallower net. Figure 6(b) visualizes the trajectory corresponding to row "2-4-3" in Table 3, which is much smoother compared to Figure 6(a). Figure 6(c)(d) visualize the trajectories of *p-AutoGrow* with $K = 50$ and 3. The 2D projection gives limited information to reveal the advantages of *p-AutoGrow*

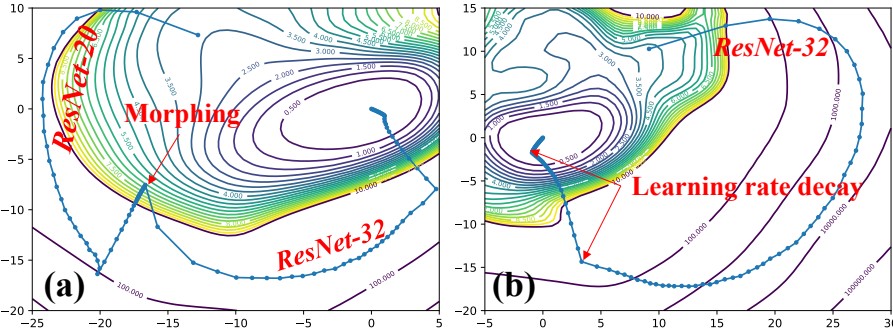

Figure 5: An optimization trajectory comparison between (a) Network Morphism and (b) training from scratch.

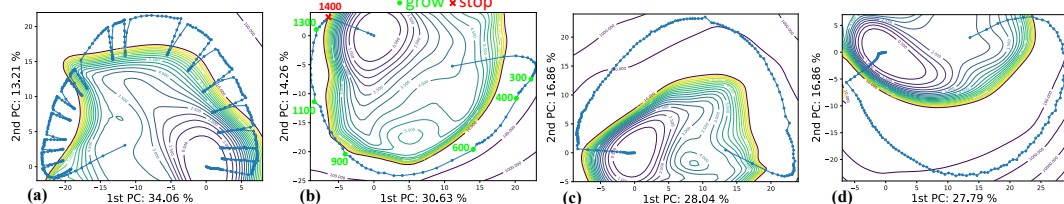

Figure 6: Optimization trajectory of *AutoGrow*, tested by `Basic3ResNet` on CIFAR10. (a) *c-AutoGrow* with staircase learning rate and `ZeroInit` during growing; (b) *c-AutoGrow* with constant learning rate and `GauInit` during growing; (c) *p-AutoGrow* with $K = 50$; and (d) *p-AutoGrow* with $K = 3$. For better illustration, the dots on the trajectory are plotted every 4, 20, 5 and 3 epochs in (a-d), respectively.

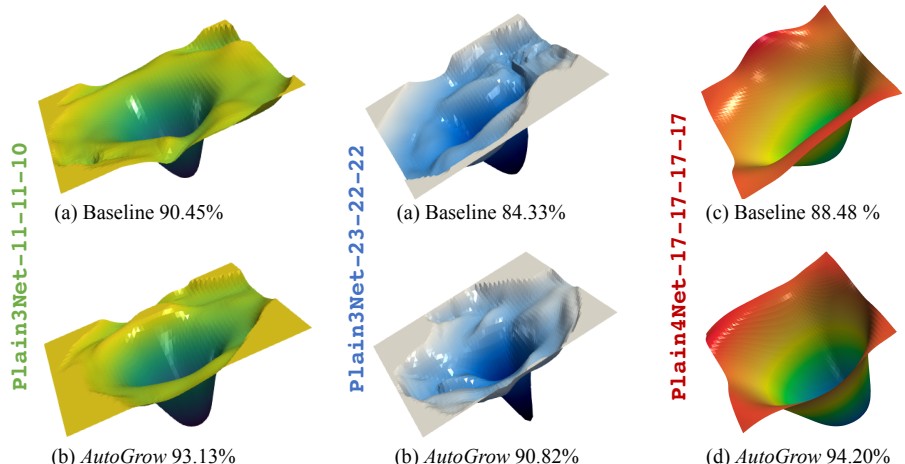

Figure 7: Loss surfaces around minima found by baselines and *AutoGrow*. Dataset is CIFAR10.

comparing to *c-AutoGrow* in Figure 6(b), although the trajectory of our final *p-AutoGrow* in Figure 6(d) is plausibly more similar to the one of training from scratch in Figure 5(b).

### A.0.2 VISUALIZATION OF LOSS SURFACES AROUND MINIMA

Figure 7 visualizes loss surfaces around minima by *AutoGrow* and baseline. Intuitively, *AutoGrow* finds wider or deeper minima with less chaotic landscapes.

### A.0.3 MORE EXPERIMENTAL RESULTS

Figure 8 plots the growing and converging curves for two DNNs in Table 10.

Table 11 summarizes the adaptability of *AutoGrow* to the sizes of dataset. In each set of experiments, dataset is randomly down-sampled to 100%, 75%, 50% and 25%. For a fair comparison, $K$ is divided by the percentage of dataset such that the number of mini-batches between growths remains

Table 8: *p-AutoGrow* under initializers with $K = 3$.

| CIFAR10 | | | CIFAR100 | | |
|---|---|---|---|---|---|
| initializer | found net[†] | accu | initializer | found net[†] | accu |
| ZeroInit | 31-30-30 | 93.57 | ZeroInit | 26-25-25 | 73.45 |
| AdamInit | 37-37-36 | 93.79 | AdamInit | 27-27-27 | 73.92 |
| UniInit | 28-28-28 | 93.82 | UniInit | 41-41-41 | 74.31 |
| **GauInit** | **42-42-42** | **94.27** | **GauInit** | **54-53-53** | **74.72** |

[†] `Basic3ResNet`          [†] `Basic3ResNet`

Table 9: *AutoGrow* improves accuracy of plain nets.

| dataset | net | layer # | method | accu % |
|---|---|---|---|---|
| CIFAR10 | `Plain4Net-6-6-6-6` | 26 | baseline | 93.90 |
| | `Plain4Net-6-6-6-6` | 26 | *AutoGrow* $K = 30$ | 95.17 |
| | `Basic4ResNet-3-3-3-3` | 26 | baseline | 95.33 |
| CIFAR10 | `Plain3Net-11-11-10` | 34 | baseline | 90.45 |
| | `Plain3Net-11-11-10` | 34 | *AutoGrow* $K = 50$ | 93.13 |
| | `Basic3ResNet-6-6-5` | 36 | baseline | 93.60 |

Table 10: The efficiency of *AutoGrow*

| net | GPUs | growing | fine-tuning |
|---|---|---|---|
| `Basic4ResNet-12-12-11-11` | 4 GTX 1080 Ti | 56.7 hours | 157.9 hours |
| `Basic4ResNet-9-3-6-4` | 4 GTX 1080 | 47.9 hours | 65.8 hours |
| `Bottleneck4ResNet-6-6-6-17` | 4 TITAN V | 45.3 hours | 114.0 hours |
| `Bottleneck4ResNet-6-7-3-9` | 4 TITAN V | 61.6 hours | 78.6 hours |
| `Plain4Net-6-6-6-6` | 4 GTX 1080 Ti | 11.7 hours | 29.7 hours |
| `Plain4Net-5-5-5-4` | 4 GTX 1080 Ti | 25.6 hours | 25.3 hours |

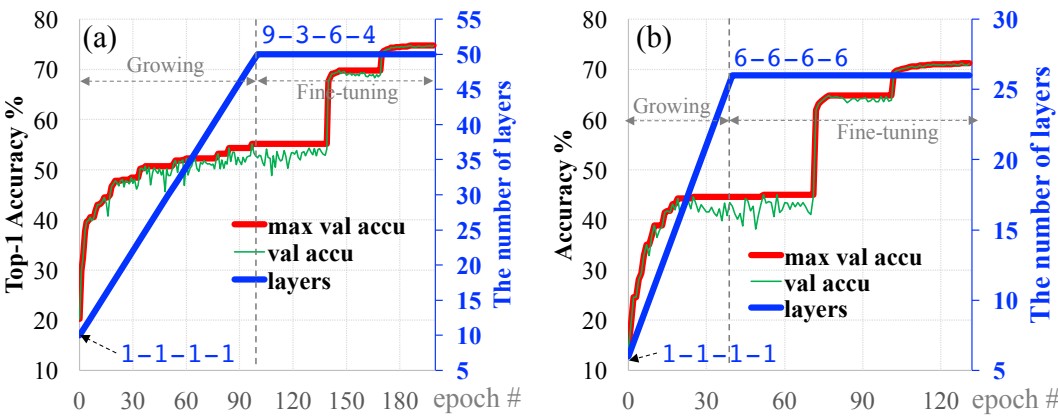

Figure 8: The convergence curves and growing process on ImageNet for (a) `Basic4ResNet-9-3-6-4` and (b) `Plain4Net-6-6-6-6` in Table 10.

Table 11: The adaptability of *AutoGrow* to dataset sizes

| `Basic3ResNet` on CIFAR10 | | | `Plain3Net` on MNIST | | |
|---|---|---|---|---|---|
| dataset size | found net | accu % | dataset size | found net | accu % |
| 100% | 42-42-42 | 94.27 | 100% | 20-20-20 | 99.66 |
| 75% | 32-31-31 | 93.54 | 75% | 12-12-12 | 99.54 |
| 50% | 17-17-17 | 91.34 | 50% | 12-11-11 | 99.46 |
| 25% | 21-12-7 | 88.18 | 25% | 10-9-9 | 99.33 |
| `Basic4ResNet` on CIFAR100 | | | `Plain4Net` on SVHN | | |
| dataset size | found net | accu % | dataset size | found net | accu % |
| 100% | 17-51-16-16 | 79.47 | 100% | 12-12-12-11 | 97.08 |
| 75% | 17-17-16-16 | 77.26 | 75% | 9-9-9-9 | 96.71 |
| 50% | 12-12-12-11 | 72.91 | 50% | 8-8-8-8 | 96.37 |
| 25% | 6-6-6-6 | 62.53 | 25% | 5-5-5-5 | 95.68 |

the same. As expected, our experiments show that *AutoGrow* adapts to shallower networks when the sizes are smaller.

