# OpenReview forum: "AutoGrow: Automatic Layer Growing in Deep Convolutional Networks"
_ICLR.cc/2020/Conference — Reject_

### Official Review · AnonReviewer2 · 2019-10-14
**Official Blind Review #2**

**Rating:** 3

**Review:**

Contributions:
	This paper best fits in the literature that explores growing network depth.  The main framework here is to interleave training a shallower network and adding new layers.  This paper (their final algorithm) differs from existing methods in that they: 1) initialize the new layers using standard initialization as opposed to the commonly used zero-init in this literature, 2) grows at a fixed interval , and this interval is short (to avoid the shallower nets being  overly-trained)., 3) uses a large and constant learning rate during the growing phase.
Empirically, they show competitive results on standard image benchmarks.
More interestingly (to me), they provide interesting insights to this paradigm of ‘growing networks’.

Comments/Questions:
Section 2 of the paper describes the proposed method is good details.

Section 3 of the paper describes the experiments.  Since for now I see the contribution of this paper is mostly empirical, I will give my detailed feedback here.
3.1 (Suboptimum of Network Morphism (NM) )
Table 2 shows NM is worse than training from scratch, and this isn’t fixed by AdamInit.
Table 3 shows c-AutoGrow (in between p-AutoGrow and NM) still does worse than from scratch, pinpoint the problem to converged subnetworks.
3.2  (p-AutoGrow)
Table 3 shows +Constant LR helps, then +RandomInit helps.
Table 4, 5 shows +Periodic gets the best performance.
*Suggestion* The found net is Table 4,5  are significantly deeper than those in Table 2,3, also there are no \Delta.  Also, although within this write-up those are the highest numbers, in the broader literature of NAS this doesn’t seem to be that good.  From a quick search, many methods in the Table 1 of [1] seems to give >96% accuracy on CIFAR10, some even close to 98%.  It might be good to at least discuss why this method is limited from achieving that.
I do like the finding that ZeroInit is unnecessary, as reported in the rest of this subsection.  However, it is unsatisfying to me that many past works (as cited by the authors) required this ZeroInit without ever trying RandomInit.
*Suggestion* I would love to see a more thorough discussion on why GauInit is better than ZeroInit, not just more numbers.  For example, even just text description on why past works found ZeroInit useful, and countering some of those claims would be interesting.  A more controlled experiment rather than training 2 networks by swapping this would be interesting.  ZeroInit is used in more context than just NM.  For example, good flow models like Glow also uses such initialization, for likely a different reason, but I wonder if findings here have any implication for ZeroInit more generally.

3.3 (Many datasets)
Table 6 is a strong result.  One odd thing is how deep the found-net has to be for MNIST.  This actually suggest to me that AutoGrow does not have the ability to stop early when it can.  And in the discussion, the authors argue that by using a better sub-module like in NAS they can do better.  This raises the question why the authors did not choose to use it.  I would believe it if the proposed method has obvious reasons that it can transfer to different architecture, but for now I cannot jump to the conclusion that, say, p-AutoGrow with GauInit will necessarily work when using a different sub-module.  Perhaps, the reason past NM works didn’t use a GauInit was also due to the fact that past sub-modules didn’t work with GauInit.

3.4 (Scale to ImageNet) It’d be good to add reference results from other papers.

Minor details:


There are some good contents in this work, but for it to be a strong *empirical* contribution, perhaps it would be more useful to include experiments on other data modality where things are not so well tuned, and show state-of-the-art results.  For it to be a strong *analysis* paper, it should expanded, at least addressing some of the *suggestions* mentioned above.
Unrelated to my evaluation of this work, reading this makes me think we should (and can) develop theoretical understanding to this paradigm of growing networks.



References:
[1] https://arxiv.org/pdf/1905.13360.pdf



**Experience Assessment:**

I do not know much about this area.

**Review Assessment: Checking Correctness Of Derivations And Theory:**

N/A

**Review Assessment: Checking Correctness Of Experiments:**

I assessed the sensibility of the experiments.

**Review Assessment: Thoroughness In Paper Reading:**

I read the paper thoroughly.

---

> ### Author Response · Authors · 2019-11-15
> **Reply**
>
> Thanks for reviewing.
>
> Replies to each item:
> *Suggestion* 1:
> * We will discuss those aspects in a revision. This paper is not a "result" paper to show a new STOA accuracy. We propose a framework that can discover an optimal depth.  "The final accuracy is limited by the sub-module design, not by our AutoGrow." Our goal in this paper is that, given a sub-module architecture, how can we find a near-optimal depth. We have briefly discussed it, but will dive deeper.
>
> *Suggestion* 2:
> * In this paper, the major difference of applying ZeroInit is that we first train a shallower net to some extent (not fully converged) and then keep adding some new layers, which are initialized by ZeroInit; while previous literature just uses ZeroInit as parameter initialization and then train it without changing the neural architecture. The ZeroInit plays different roles:
> ** in previous literature, an intrinsic shallower net is not trained (not converged) at the beginning and, after just few iterations, the ZeroInit layers turn to non-zeros (random values) quickly, which is why those work well with ZeroInit. We argue that the previous work is more like a random initialization after training for a few iterations.
> ** in our literature, the shallower net is trained to some extent; using ZeroInit to initialize new layers will stick to local minimum. This is proved in many experiments in this paper.
> In summary, the settings in this paper and in previous ones are different, we didn't argue that ZeroInit will be worse than random initialization in the settings that the reviewer mentioned.
> We will add this discussion in a revision.
>
> * "(Many datasets)"
> ** We showed in Figure 3 that AutoGrow can stop at a reasonable depth, and the depth found in ImageNet is also reasonable. We indeed observe a very deep net for small datasets (such as MNIST), and we are integrating pruning into the growing to tackle this issue. We hope this paper is the first step to achieve an ideal goal for all settings.
> ** we had limited time to duplicate and run results using NASNET cell. The work is on-going.
>
>
> * "(Scale to ImageNet)"
> We will. One method that *directly* searches in imagenet is [1], which achieved "75.2 ± 0.4%" top-1. This value is lower than our "77.99%". The fact questions if we really need to search over all the combinations or searching depth will be sufficient.
>
> * Mirror
> Agree. We wish our empirical results can inspire "theoretical understanding" in the future. Growing architecture is challenging and new, and we wish our work can inspire others.
>
> [1] Bender, Gabriel, Pieter-Jan Kindermans, Barret Zoph, Vijay Vasudevan, and Quoc Le. "Understanding and simplifying one-shot architecture search." In International Conference on Machine Learning, pp. 549-558. 2018.

---

### Official Review · AnonReviewer3 · 2019-10-24
**Official Blind Review #3**

**Rating:** 3

**Review:**

This paper's contribution is a method for automatically growing the depth of a neural network during training. It compares several heuristics that may be used to successfully achieve this goal and identifies a set of choices that work well together on multiple datasets.

The paper focuses on CNNs that conform to a popular design pattern where the network is organized into a series of sub-networks, each consisting of a series of sub-modules (sometimes called blocks) operating at the same resolution. To be precise, the proposed method aims to learn the length of each series of sub-modules. A main contribution of the paper is the demonstration that it is not necessary to train a network until convergence before adding new sub-modules as proposed in past work. Instead, it is better to grow the network after training for a short while.

My current decision for this paper is a weak rejection due to the points below. However, I am open to revising my opinion if these points are addressed satisfactorily.

- The growing strategy identified in the paper as a superior alternative seems to be already known and used, at least in the speech recognition community. Seide et al. (2011) called it Discriminative Pre-training, and showed that it outperforms greedy layer-wise pretraining and DBN pre-training. Zeyer et al. (2017) reported that a similar method also enables the training of very deep LSTM networks which is otherwise notoriously hard. In general, the existence of prior work with the same ideas does not preclude acceptance, but the existence of this work needs to be clearly stated early on and the additional value of the current study sufficiently clarified.

- I find it strange that the final networks found by the proposed method usually have the same/similar number of sub-modules per sub-network (Tables 4,5,6) on multiple datasets. The only exceptions appear to be Basic4ResNet/CIFAR100 in Table 6 and about 50% of ImageNet results in Table 7. This regularity suggests that either A) the proposed algorithm prefers to set same number of sub-modules per sub-network due to its design, or B) datasets except ImageNet have an inherent shared property that produces this result. Since option A suggests a bias in the algorithm, this peculiarity of the results needs to be investigated or explained further.

- Figure 3 constitutes the main evidence that Autogrow finds approximately optimal depths as compared to manual searching, but it is not clear how the plot for baselines is obtained. For any given parameter budget, there are multiple baseline networks possible since the sub-networks can have different number of sub-modules (see previous point). This does not appear to be accounted for in Figure 3. Further, when dealing with CNNs, it would be more useful to have computation budget on the x-axis instead of the parameter budget. This would better account for the difference between increasing depth in an earlier sub-network vs. a later one.

- The reported results appear to be for single trials throughout the paper. This does not seem sufficient especially for results in Tables 2 and 3 where many differences are rather small, and so drawing conclusions from these tables would be unscientific.

References:

Seide, Frank, et al. "Feature engineering in context-dependent deep neural networks for conversational speech transcription." 2011 IEEE Workshop on Automatic Speech Recognition & Understanding. IEEE, 2011. https://www.microsoft.com/en-us/research/wp-content/uploads/2016/02/FeatureEngineeringInCD-DNN-ASRU2011-pub.pdf

Zeyer, Albert, et al. "A comprehensive study of deep bidirectional LSTM RNNs for acoustic modeling in speech recognition." 2017 IEEE International Conference on Acoustics, Speech and Signal Processing (ICASSP). IEEE, 2017. https://arxiv.org/abs/1606.06871

Update after rebuttal
-----------------------------
I'm sympathetic to the unfortunate situation that the authors are in, since the underlying growing strategy has already been covered by prior work. As I mentioned earlier, a sufficient rewrite of the paper can clearly state what has been done already so as not to take credit from the earlier authors. A revised version of the paper has not been uploaded; I suggest that the authors do so for the future.

I agree that the focus of this paper is learning the 'optimal' depth by using the growing strategy. But I am not convinced that the technique indeed finds optimal depths based on the regularity of the sub-network depths mentioned in my review. The rebuttal suggests reasons for the obtained regularity, but does not prove that these regular structures obtained are indeed optimal and not an artifact of the algorithm itself. The baselines are also using the same regular architectures, which distorts the overall picture because it is possible that a non-regular architecture provides a better trade-off.

While my rating doesn't change, I do think that the work is in an interesting direction. My final suggestions for the future are:
- Investigate where non-regular architectures (unequal sub-network depths) are in the trade-off between accuracy, flops and parameters.
- Investigate whether the proposed algorithm can be modified to easily find non-regular architectures if they can yield equally good performance as regular ones at similar or lower cost.


**Experience Assessment:**

I have read many papers in this area.

**Review Assessment: Checking Correctness Of Derivations And Theory:**

N/A

**Review Assessment: Checking Correctness Of Experiments:**

I carefully checked the experiments.

**Review Assessment: Thoroughness In Paper Reading:**

I read the paper thoroughly.

---

> ### Author Response · Authors · 2019-11-15
> **Reply**
>
> Thanks for reviewing.
>
> Responses to each item:
> - Thanks for pointing us to more related works. We will include them in this paper ASAP. The similar research in speech recognition community proves that our approach can be generic to other domains, and our research also indicates that their approaches can potentially generalize to computer vision domain. We quite appreciate similar discoveries of "(1) stopping very early by going through the data only once and (2) using large learning rate" in that paper. The major difference in this paper is that we are using AutoGrow to find an optimal depth, instead of enabling training deeper networks, which were hard to train but are easily now because of many recent advances such as weight initialization, batch normalization, shortcut paths, just to name a few.
>
> - We believe that the "regularity" could be because of both. We believe that this is because of the nature of a periodic growing with a short interval and because of the small size of dataset. AutoGrow will only stop when the whole network saturates in that case. While in ImageNet, the dataset is larger, therefore, the "regularity" is infrequent.
>
> - For baselines, we set the numbers of sub-modules (under each resolution) the same (such as K), then we sweep K from 1 to a large number. We list the baselines for Basic3ResNet below and we will include all others in a revision. We also include the computation in "Flops" below as requested. We used "Params" as we think the number of layers is more related to the number of parameters, but we will include both in a revision.
> -----------------------------------------------------------------------------
> Net                                                 | Flops            | Params       | Accuracy
> Basic3ResNet-[3, 3, 3]           41214592       272474            92.96
> Basic3ResNet-[5, 5, 5]          69755520        466906            93.44
> Basic3ResNet-[7, 7, 7]          98296448        661338            93.68
> Basic3ResNet-[9, 9, 9]          126837376      855770            93.88
> Basic3ResNet-[24, 24, 24]    340894336      2314010          94.26
> Basic3ResNet-[42, 42, 42]    597762688      4063898          94.3
> Basic3ResNet-[72, 72, 72]    1025876608    6980378          94.17
> -----------------------------------------------------------------------------
>
> - The conclusion holds for each run. We treat both ZeroInit and AdamInit as network morphism and, in table 2, we concluded that ZeroInit and AdamInit were similar and sub-optimal.
> We run the the first setting in Table 2 for three runs, and the accuracy is 92.73%, 92.86%, 92.51%.
> For table 3, we run one more "CIFAR100 constant GauInit", we get 70.83%, and one more "CIFAR100 staircase ZeroInit" and get 70.11%.

---

### Official Review · AnonReviewer1 · 2019-10-27
**Official Blind Review #1**

**Rating:** 3

**Review:**

The paper presents a meta-learning algorithm to automatically detemine the depth of neural network through a policy to add depth if this bring improvement on accuracy.

I have conserved opinion based on the technique being used here is extremely simple, basically is an implementation of naive greedy algorithm in such a scenario, which implies the problem may not be intrinsically hard, or even useful. The paper consists of detailed narrative about how these procedure are conducted, but still, it is really hard for me to find the true merit to appreciate, and why this brings a nontrivial and usefull contribution. The tables, visualization figures also didnot imply too much about whether this is more than overfitting on previous works with hand-chosen depth.

**Experience Assessment:**

I have published one or two papers in this area.

**Review Assessment: Checking Correctness Of Derivations And Theory:**

I assessed the sensibility of the derivations and theory.

**Review Assessment: Checking Correctness Of Experiments:**

I assessed the sensibility of the experiments.

**Review Assessment: Thoroughness In Paper Reading:**

I read the paper at least twice and used my best judgement in assessing the paper.

---

> ### Author Response · Authors · 2019-11-15
> **We should prefer simple and worked methods, instead of complex and hard-to-use methods**
>
> Thanks for the review.
>
> We actually started from more complex and "nicer" methods, such as network morphism.
> However, our experiments showed that those more complex methods worked worse than our simple solution (covered in this paper).
> Therefore, one of our contribution is that complex growing is NOT necessary, a simple periodic  growing can even does better.
>
> Then the question is: should we appreciate more complex methods which have worse results or appreciate simple but more effective methods?
>
> Others:
> * "implies the problem may not be intrinsically hard, or even useful.": growing architecture is harder than pruning architecture [1]. Pruning starts from a predefined large space and just need to find a subspace within it, while growing performs in a reversed order, during which the search space is totally open. A simple pruning method [1] inspires lots of following work and we wish our method can inspire others from a new perspective. We also believe, in the future, that combining pruning and growing will be important.
> We need more details from the reviewer regarding why automatically growing neural architecture will not be useful.
> * "detailed narrative" is exactly how we showed that complex methods are not necessary and can even be worse.
> * "overfitting": all results are cross validated by a standard machine learning procedure. It is not a question specifically for our paper. It is a question for the whole community that if using validation dataset of ImageNet can result in overfitting.
>
> [1] Han, Song, Jeff Pool, John Tran, and William Dally. "Learning both weights and connections for efficient neural network." In Advances in neural information processing systems, pp. 1135-1143. 2015.

---

### Author Response · Authors · 2020-06-11
**Updated version in KDD 2020**

Thank all comments! An updated version in KDD 2020 is at https://arxiv.org/abs/1906.02909

---

### Decision · Program_Chairs · 2019-12-19

**Decision:**

Reject

**Comment:**

This paper proposes a neural architecture search method based on greedily adding layers with random initializations. The reviewers all recommend rejection due to various concerns about the significance of the contribution, novelty, and experimental design. They checked the author response and maintained their ratings.